# The miRNAs Role in Melanoma and in Its Resistance to Therapy

**DOI:** 10.3390/ijms21030878

**Published:** 2020-01-29

**Authors:** Francesca Varrone, Emilia Caputo

**Affiliations:** 1IRBM S.p.A., Via Pontina Km 30,600, I-00071 Pomezia, Italy; 2Institute of Genetics and Biophysics –I.G.B., Adriano Buzzati-Traverso, Consiglio Nazionale delle Ricerche (CNR), Via Pietro Castellino, 111, I-80131 Naples, Italy

**Keywords:** melanoma, metastasis, drug resistance, miRNAs

## Abstract

Melanoma is the less common but the most malignant skin cancer. Since the survival rate of melanoma metastasis is about 10–15%, many different studies have been carried out in order to find a more effective treatment. Although the development of target-based therapies and immunotherapeutic strategies has improved chances for patient survival, melanoma treatment still remains a big challenge for oncologists. Here, we collect recent data about the emerging role of melanoma-associated microRNAs (miRNAs) currently available treatments, and their involvement in drug resistance. We also reviewed miRNAs as prognostic factors, because of their chemical stability and resistance to RNase activity, in melanoma progression. Moreover, despite miRNAs being considered small conserved regulators with the limitation of target specificity, we outline the dual role of melanoma-associated miRNAs, as oncogenic and/or tumor suppressive factors, compared to other tumors.

## 1. Introduction

### 1.1. Melanoma: Its Progression and Metastasis

Melanoma is the most malignant skin cancer and its incidence has been steadily increasing over the past three decades, accounting for the majority of skin cancer-related deaths worldwide [1]. Melanoma originates from an altered proliferation of melanocytes, a skin minority cell population, with a low proliferative potential, deriving from neural crest cell precursors. They produce melanin pigment, providing it to the nearby keratinocytes. UV exposure induces melanocytes to proliferate and produce melanin, by a finely tuned process, depending upon specific pathways of which alteration is associated to the melanocyte malignant transformation [2,3].

Melanomagenesis has been classically described as a process characterized by a linear progression of normal melanocytes through various precursor lesions and ultimately to melanoma [4]. The initial step is the formation of a nevus, which consists of the proliferation and aggregation of melanocytes located at the skin layer and epidermis; some of them could also show an altered growth pattern changing into a dysplastic nevus phenotype. However, it may arise from a preexisting melanocytic nevus or as a new lesion. This stage of progression is characterized by the disruption of the p16^INK4a^-retinoblastoma (Rb) pathway, mostly by inactivation of CDKN2A, due to its mutations [5,6]. Successively, the continuous and unchecked melanocyte proliferation allows them to penetrate into the epidermal/dermal junction or within the dermis by a radial growth phase (RGP). At this stage, neoplastic melanocytes show an immortal phenotype, achieved by the activation of human telomerase reverse transcriptase (hTERT) [7]. Then, cells may grow into the dermis, where, they interact with other cell types and gain physical access to both lymphatics and blood vessels, and may enter into the final step of progression [4]. The final stage of melanoma progression is characterized by a vertical growth phase (VGP), requiring mutations repressing apoptosis, which allow cells to survive in the absence of keratinocytes as well as PTEN loss, over-expression of various protein kinases or RAS activation, and β-catenin activation (reviewed in Bennett et al.) [8]. The loss of E-cadherin along with the aberrant expression of N-cadherin and αVβ3 integrin have also been shown to be crucial for the progression from RGP to VGP final melanoma progression (reviewed in Miller et al.) [9] (Figure 1).

Although Clark classification is still used for melanoma stadiation, substantial evidence suggests that melanomagenesis is not a result of a linear progression of alterations from nevus through all the different phases to the metastatic melanoma, but MM can originate from each of the described phases, without necessarily passing through all of them. 

In fact, this model has been improved from the discovery and identification of cancer stem cells (CSCs). It has been demonstrated that malignant melanoma stem cells (MMSCs) are involved in the melanoma origin, progression, and metastasis [11]. MMSCs are similar to adult stem cells for their multipotency and plastic phenotype properties [12,13,14]. These cells express VE-cadherin, and the receptor tyrosine kinase for ephrin (Eph), and are able to induce de novo tumor angiogenesis by a vasculogenic mimicry (VM) process. At the same time, the growth factors produced from endothelial cells and/or fibroblasts present in the microenvironment play an important role in melanoma progression, as well as immune cells infiltrating the tumor mass and their cytokines [15,16,17,18]. 

Furthermore, our growing knowledge about the epigenetic mechanisms involved in cancer development provide a more complex picture of melanoma progression and metastasis [19,20].

### 1.2. Melanoma Risk Factors

Melanoma progression is a combination of environmental and genetic risk factors [21,22]. Among the environmental factors, the accumulation of sun exposure (or by UV radiation from tanning beds) is the primary and most common risk factor. Melanocytes are resistant to UV-induced apoptosis. They continue to grow and accumulate genetic mutations, leading to the formation and growth of a melanoma [23,24]. Other environmental risk factors for melanoma have been demonstrated to be the skin pigmentation phenotype (fair or light skin) as well as the existence of atypical and multiple nevi [25,26,27,28]. A role of the circadian rhythm in melanoma development has also been reported. It has been demonstrated that melatonin, a major output product of thee circadian rhythm, plays a protective role in melanoma [29].

Along with the environmental risks, genetic risk factors have a large impact on melanoma onset and progression. Somatic mutations in genes, critical in the regulation of pathways, involved in cell proliferation, differentiation, and survival have been identified in approximately 60% of melanomas. (e.g., BRAF, NRAS, TP53, NF1, MITF, c-kit, CDKN2A, PTEN, RAC1, GNAQ, GNA11, and RPS27) [30,31,32,33]. 

Although sporadic melanoma is the most common, melanoma may occur in multiple members of the same family and it may be due to shared environmental risk factors and/or genetic mutations, or both [34,35,36,37].

### 1.3. Melanoma Current Treatment 

Surgery represents the principal treatment for accessible and early stage cutaneous melanoma (I–II stage). Although recent progresses in melanoma diagnosis and treatment have contributed to substantially increasing the metastatic cancer patient survival, treating metastatic melanoma (MM) still remains a key challenge for oncologists [38,39]. 

Here, we briefly summarize the main categories of current melanoma treatment:

#### 1.3.1. Surgical Treatment

Surgery, as mentioned above, is used for early melanoma (I–II stage) treatment. After complete excision of a primary melanoma radiation therapy (RT), can be considered as adjuvant therapy [40]. 

Because of the high number of metastases or the low accessibility and difficult of detecting small metastatic lesions by using current imaging tools, MM is highly unlikely to be cured by surgery. Despite this, it has been demonstrated that some melanomas in stage III (rarely IV), having not been widely spread and carrying the BRAF mutation, can be treated surgically. In these cases, it has been demonstrated that their treatment with the BRAF/MEK inhibitors combination, as a neoadjuvant and adjuvant before and after surgery, provided a significant change in preventing later recurrence [41].

#### 1.3.2. Chemotherapy

For melanoma patients with progressive, refractory, or relapsed disease, chemotherapy has been used for over three decades. The most commonly used chemotherapeutic agents have been temozolomide (TMZ) and dacarbazine (DTIC), alkylating agents, which are able to inhibit DNA synthesis by stopping or slowing the growth of cancer cells [42,43]. 

DTIC has been the standard care until the approval of the first target-based therapy with vemurafenib in 2011, a BRAF inhibitor (BRAFi), along with ipilimumab, the first immune checkpoint inhibitor, a cytotoxic T-lymphocyte-associated protein 4 (CTLA-4) antibody. 

Since then, several target-based therapeutics and immune checkpoint inhibitors have been approved for MM treatment. Therapies based on these agents have significantly increased the median overall survival (OS) of MM patients from ~9 months before 2011 to 2 plus years, and, in some cases, producing long-term disease remissions. 

Nowadays, chemotherapy is only used after and when more effective treatments, such as target therapy and checkpoint blockade treatments, fail.

#### 1.3.3. Target-Based Therapy 

It has been shown that the mitogen-activated protein kinase (MAPK) pathway is critically involved in melanoma pathogenesis [44]. Some of the genes encoding the proteins belonging to the MAPK cascade are mutated in melanoma cells, compared to melanocytes, providing a solid strategy for the development of target-based therapies. 

About 40 different mutations in the BRAF gene have been detected. The most frequent has been found in exon 15 and it is responsible for substitution of valine in glutamic acid at position 600 of the BRAF protein (BRAF-V600E-) [45,46]. It constitutes about 90% of the BRAF mutations observed in melanoma and in almost 50% of melanoma patients [47]. The mutated BRAF gene encodes for an active BRAF protein inducing the constitutive MAPK pathway activation and subsequently promoting cell proliferation and preventing apoptosis in melanoma cells [48].

Other mutations have been also identified in the NRAS gene in approximately 15–20% of melanomas [44,49,50].

Furthermore, molecular alterations have also been identified in other genes, such as c-KIT and GNαQ (Guanine Nucleotide-binding protein G(q) subunit alpha (Gαq)), which have been described in mucosal and uveal melanoma subtypes, respectively [51].

Based on these findings, several agents inhibiting the mutated BRAF and MEK proteins have been developed and, since 2011, different target-based therapeutics have been approved by the Food & Drug Administration (FDA), as shown in Figure 2A. 

However, recent investigations have shown that the continuous treatment with BRAF and MEK inhibitor agents (BRAFi, MEKi) of patients carrying BRAF-mutant melanoma consistently failed due to the selection of genetic mutations conferring disease resistance or the ability of melanoma endorsing drug resistance-associated transcriptional programs [52,53].

Since 2014, the combination of two drugs has been extensively studied in order to escape the phenomenon of drug resistance [54,55,56]. It has been demonstrated that intermittent BRAFi dosing, using fixed on/off schedules, leads to a resistance time delay in preclinical studies [54]. 

Furthermore, patients with BRAF-mutant melanoma now have a new treatment option: Encorafenib (a new generation BRAFi) with binimetinib (a new generation MEKi). The duration of response to this treatment in a pivotal trial was 16.6 months and responses were seen in 63% of patients. Compared to the previously FDA-approved combination of dabrafenib (BRAFi) and trametinib (D/T), the new drugs appear to have similar response rates but a longer duration of response [57]. However, a direct comparison of the two combinations in a clinical trial will be needed to make a solid conclusion about the comparative efficacy of the two treatments [56]. 

#### 1.3.4. Immunotherapy 

Several different types of immunotherapy have been developed over the years, as illustrated in Figure 2B. The most important are checkpoint inhibitors that stimulate or trigger the immune system to attack and kill the cancer cells. 

Early immunotherapeutic agents, interleukin-2 (IL-2) and interferon α-2b (IF-α 2b), a recombinant analogue, were FDA approved as single agents in 1998 and 2011, respectively. These agents act by stimulating the immune system activity against the cancer and have been used in resected stage II/III patients and sometimes in stage IV melanoma patients. In these cases, an overall response rate of 10–20% with a small effect on survival and a durable long-term benefit in less than 10% of patients was observed [58]. 

However, an upregulation in CTLA-4 and PD-L1 expression in malignant cells, as well as a downregulation of the antigen presentation processing has been demonstrated, depending on their altered genetic and epigenetic properties [59]. These cells are also able to adopt alternative signaling pathways to prevent cytotoxic T cells from causing cancer cell death silencing the immune surveillance and growing unchecked. 

Ipilimumab, an inhibitor of CTLA-4 (anti-CTLA-4), has been approved for the treatment of advanced or unresectable melanoma This agent has been shown to be effective as monotherapy and in combination with nivolumab (anti-PD-1) [60]. Another antibody, anti-PD-1, is pembrolizumab, and is FDA approved for melanoma and the treatment of different types of cancer. These agents show a dramatic increase in the durable response rates and a manageable safety profile in monotherapy. However, more than 50% of patients did not respond to this treatment. 

In order to enhance the response rates in patients, combined drugs were also evaluated. In particular, the combination of ipilimumab (anti-CTLA-4) plus nivolumab (anti-PD-1) has been observed to be significantly more efficient in metastatic melanoma patients compared to their use in monotherapy [61]. 

Furthermore, oncolytic virus anti-cancer therapy has recently been evaluated for melanoma treatment. This therapeutic strategy relies on the oncolytic virus’ ability of indirectly lysing tumor cells, leading to the release of soluble antigens and interferons, driving the antitumor immunity. In particular, the attenuated herpes simplex virus-based oncolytic virus talimogene laherparepvec (T-VEC) was FDA approved in 2015, and it is currently used as a local treatment of patients carrying an unresectable advanced stage melanoma [62]. 

Recently, TLR9 agonists (SD-101, CMP-001, IMO-2125) were developed as new immunotherapeutics, which are intratumorally injected mostly in combination with other drugs as a pairing therapy [63]. Scientific investigations have shown their ability to increase the expression of TRL9 receptor, a critical key factor in the activation of the innate and adaptive immunosystem in cancer [64]. Interestingly, Milhem et al. also reported a positive response to these drugs in the tumor sites not locally injected with the drug (abscopal effect) [65].

#### 1.3.5. Vaccine

Tumor cells are characterized by genetic instability, resulting in the occurrence of a large number of mutations, as well as the expression of non-synonymous mutations producing tumor-specific antigens, also known as neo-antigens. Researchers from Dana-Farber Cancer Institute, the Broad Institute of MIT, and Harvard created vaccines targeting tumors’ neoantigens. These molecules are new to the immune system and may trigger an immune response. It has been reported that in patients with melanoma, a personalized treatment vaccine generated a robust immune response against the cancer and may have helped to prevent it from returning [66]. Although this clinical trial is in the I phase, the studies confirm the potential of neo-antigen vaccines to treat cancer and should lead to larger trials in the future in order to help address such challenges for effective cancer immunotherapy.

#### 1.3.6. Pairing Therapies

It has been demonstrated that the combination therapy, among the target-based therapy and immunotherapy, is more efficient than monotherapy. A combination of immunotherapy drugs for people who have less advanced melanoma was evaluated by Heinzerling et al. They showed the efficiency of BRAF and MEK inhibitors (BRAFi + MEKi), combined, as an established therapeutic option in patients with BRAF-mutated advanced melanoma. In particular, the dabrafenib + trametinib (D + T) combination has been demonstrated to prolong overall survival in the adjuvant setting [67]. 

At the same time, it seems that the combination of ipilimumab and nivolumab (CTLA-4 + PD-1 antibodies) is preferred to the mono-treatment in immunotherapy. Abdel-Wahab et al. demonstrated that the combination therapy CTLA-4 + PD-1 was more effective than previous monotherapy regimens in small patient cohorts selected for the first phase I trials, although it increased drug-related toxicity [68]. These data were also confirmed in phase III trials, supporting the use of anti-CTLA-4 and anti-PD-1 combination therapy [69]. 

Furthermore, BRAF/MEK inhibitor combination therapy induces an elevated initial response rate with a median duration of response of approximately 1 year. The immunotherapy by targeting PD-1 produces lower response rates but a longer response duration. 

Preclinical models suggest a combination of immunotherapy and target-based therapy for patients with stage IV melanoma. BRAF and MEK inhibitors combined with anti-PD-1 agents improved antitumor activity, suggesting additional therapeutic possibilities for patients unlikely to have long-lasting responses to either mode of therapy alone [70,71]. 

The use of immunotherapy and target-based therapy has improved survival for most patients, and they are now the preferred approaches for patients with metastatic melanoma. 

Recently, a new treatment option as a target-based therapy has been developed: Encorafenib (BRAFi) plus binimetnib combination [57]. This treatment, compared to the previously FDA-approved combination (D/T), appears to have similar response rates but a longer duration. 

For immunotherapy, a bright new hope is represented by TLR9 agonists in combination with the humanized antibody pembrolizumab: SD-101, CMP-001, or IMO-2125 in combination with ipilimumab (that activates the immune system by targeting CTLA-4) [65]. However, Ribas et al. suggested that the triple-combined therapy may benefit metastatic melanoma patients carrying the BRAFV600-mutation by increasing the frequency of long-lasting anticancer responses [72]. 

Although current treatment has increased patients’ OS (overall survival), new treatment approaches are needed. Here, we will briefly describe the currently available data about the role of miRNAs in melanoma and their potential in developing innovative diagnostic tools and efficacious therapeutic drugs.

## 2. MicroRNAs in Melanoma Cell Biology

MicroRNAs (miRNAs) have been extensively studied and, since their discovery in 1993 in the *C.elegans* animal model by the Ambros’ group at Harvard University, their involvement in determining and/or repressing the tumor phenotype as well as in its prognosis and response has been well characterized [73]. 

In Table 1, we collected some of the most important miRNAs exhibiting onco-suppressor properties by targeting oncoproteins (miRNA tumor suppressor) and/or able to target mRNA-coding tumor suppressors (oncomiRs). 

It has been observed that miRNAs are involved in melanomagenesis. In particular, it has been demonstrated that mi-RNAs play an important role in MITF regulation. Microphthalmia-associated transcription factor, MITF, is a master regulator not only in melanocytes’ differentiation, proliferation, and survival but also in melanomagenesis [144]. Furthermore, it is associated to the melanoma heterogeneity. Subpopulations of cells showing different MITF cellular levels have been detected in melanoma, some showing high MITF levels, which were highly differentiated and proliferative, and others with low MITF levels, exhibiting a high invasive and metastatic potential. These findings suggested a ‘phenotype switching’ between these populations as a model to explain melanoma heterogeneity, which is the biggest issue to overcome for the development of efficacious therapeutics [145,146,147,148]. 

MITF activity is tightly modulated at the transcriptional, post-transcriptional, and post-translational levels. Several miRNAs, such as miR-137, miR-148, miR-182, miR-26a, miR-211, miR-542 3p, miR-340, miR-101, and miR218, have also been described to be involved in its regulation, as schematically shown in Figure 3.

In particular, it has been reported that miR-137 downregulates MITF expression in melanoma cell lines and its expression has been observed to correlate with the poor survival of melanoma patients at stage IV. Further, miR-137 is involved in the downregulation of multiple oncogenic target mRNAs, including c-MET (a protooncogene encoding for a tyrosine kinase receptor), YB1 (Y box-binding protein 1), EZH2 (enhancer of zeste homolog 2), and PIK3R3 (phosphatidylinositol 3 kinase regulatory 3) [82]. 

Regarding miR-137, it has been observed that miR-148 negatively regulates MITF expression in melanoma cells by targeting a binding site found in its 3′UTR sequence [84]. However, the combined miR-137 and miR-148 overexpression does not result in a cumulative effect. Interestingly, miR148 has been found to play a dual/opposite role in MITF regulation [84].

Additionally, miR-182 has been found to be frequently amplified and upregulated in melanoma cell lines as well as in tissue samples [133]. It has been observed that miR-182 overexpression stimulates the migration of, and melanoma cell survival by directly downregulating MITF and FOXO3 (forkhead box O3) expression. In particular, it has been demonstrated in A375 melanoma cell line that miR-182 overexpression induces increased proliferation, migration, and invasion, as well as inhibiting cell apoptosis, and blocking the cell cycle at the S phase [133]. 

Furthermore, miR-218 has been shown to suppress MITF expression, by targeting its mRNA 3′-UTR. It is also able to block tyrosinase synthesis and to stimulate skin melanogenesis while miR-340 drives a decrease in MITF expression and mRNA degradation because it is able to interact with two target sites on MITF 3′UTR [114,149]. 

Additionally, miR-26a and miR-101 have been demonstrated to be capable of inhibiting the invasion and proliferation of melanoma cells by targeting MITF [77,150].

Another miRNA involved in MITF regulation is miR-211. Lower miR-211 levels have been observed in highly invasive melanoma cell lines compared to less invasive ones. It has been demonstrated that miR-211 inhibits the migration and invasion of melanoma cells. It also induces the loss of cell adhesion by directly regulating NUAK1 m-RNA, an AMP-activated protein kinase-related kinase overexpressed in many cancers [151]. Also, miR-211 revealed an important role in the regulation of POU3F2, the POU domain transcription factor that is better known as BRN2, a well-established MITF repressor, suggesting a further indirect influence of miR-211 in the development of melanoma metastasis [144,146,147,152,153].

BRN2 together with MITF are key factors identified in melanoma phenotype switching: The transformation of melanocytes to malignant melanoma and the subsequent development of invasion and metastasis.

It has been reported that melanoma phenotype switching has similarities to the epithelial–mesenchymal transition (EMT) program that occurs during development and plays a critical role in the acquisition of metastatic properties during the melanoma vertical growth phase [4,154,155]. In fact, it has been demonstrated that melanoma phenotype switching is associated with MITF levels: High MITF levels have been found to be associated with the proliferative state while low MITF levels were associated with the invasive state [156]. 

Some miRNAs have been revealed to play a role in the EMT process. MiR-200c, a well-established central EMT regulator in different cancers, has been proven to be helpful in inhibiting EMT in experimental vaccination against melanoma [102,157].

Similarly, miR-542 3p is another key regulator of the EMT process. MiR-542 3p has been found to be strongly downregulated in melanoma cell lines and tissues compared to healthy counterparts. It has been demonstrated that forced miR-542 3p re-introduction inhibits the EMT process and metastasis formation in a melanoma pre-clinical model, likely by the translational inhibition of the PIM1 factor, a well-known promoter of cancer growth and spreading [118]. 

At the same time, increased expression of MITF has also been correlated with drug resistance. Ji et al.’s investigations revealed that vemurafenib (BRAFi) resistance is correlated with the loss of MITF [158].

### 2.1. MicroRNAs in in Drug Resistance

The continuous treatment of melanoma with target-based therapy leads to therapy failure due the acquisition of drug resistance. Notably, BRAFi monotherapy may provide profound initial tumor regression in patients with BRAF V600-mutated metastatic melanoma. However, this initial regression is successively followed by disease progression due to resistance establishment to the treatment. Also, the BRAF/MEK inhibitors pairing treatment often provides remarkable disease regression initially but resistance to therapy then occurs within 12 months. 

Drug resistance severely limited the efficacy of target-based therapy in BRAF-mutated metastatic melanoma. Overall, recent findings show the positive implication of miRNAs as a strategy to improve drug responses; in fact, some of them are able to provoke drug resistance and others to increase or restore drug sensitivity. 

In Table 2, we listed several miRNAs with a potential role of drug sensitivity and drug resistance to melanoma treatment with target-based therapy. 

For instance, Sun et al. showed that miR-7 could reverse resistance to BRAFi in certain vemurafenib-resistant melanoma cell lines [159]. Furthermore, miR-126 3p expression was significantly downregulated in the dabrafenib-resistant sublines as compared with their parental counterparts. It has been demonstrated that its replacement in the drug-resistant cells leads to the inhibition of proliferation, cell cycle progression, invasiveness, and increased dabrafenib sensitivity [162]. 

MiR-34a, miR-100, and miR-125b have also been shown to be highly expressed in both resistant cells and treated patient tumor biopsies. Their expression has been found to be associated to the chemokine monocyte chemoattractant protein-1 (CCL2), which promotes tumor progression in the resistant cells, suggesting that both CCL2 and miRNAs may be helpful potential prognostic factors and attractive targets for counteracting treatment resistance in metastatic melanoma [165]. 

Zheng’s study revealed that miR-31 is able to regulate resistance to chemotherapy of melanoma. MiR-31 was found to be downregulated in melanoma tissues, and its enforced expression suppressed the growth of melanoma cells and increased their chemosensitivity [160]. Also, Koez demonstrated that miR-125a inhibition induces the suppression of resistance to BRAFi in a subset of resistant melanoma cell lines, leading to a partial drug resensitization.

Finally, they showed that miR-125a upregulation is mediated by TGFβ signaling [167]. Upregulation of miR-204 5p and miR-211 5p improved vemurafenib’s responses, facilitating the emergence of resistance [168]. Moreover, miR-579 3p has been found to be downregulated in tumor samples derived from patients before and after the development of resistance to target-based therapies as well as in cell lines resistant to BRAF/MEKi [116]. 

Stark et al. demonstrated that miR-514a overexpression was correlated with increased melanoma cell resistance to BRAFi, through decreased expression of the NF1 tumor suppressor. Moreover, while miR-7, miR-34a, miR-100, and miR-125b have been shown to be able to reverse/restore melanoma resistance in target-based therapies by targeting different signaling pathways, miR-579-3p has been found to be associated with resistance development in melanoma. MiR-579-3p is observed to be downregulated in melanoma patients upon the development of resistance to target-based therapies [116,159,165,168]. 

Interestingly, miRNAs may also be associated with melanoma resistance to treatment with immune checkpoint inhibitors (ICIs). Different circulating miRNAs (let-7e, miR-99b, miR-100, miR-125a, miR-125b, miR-146a, miR-146b, and miR-155) have been demonstrated to be associated with the activity of MDSCs (myeloid-derived suppressor cells) in melanoma patients [171]. Furthermore, miRNAs have been shown to also be associated with pairing therapies involving target-based therapy. 

Wagenseller et al. detected significant upregulation of 15 miRNAs by using microarray analysis. Melanoma tissues derived from patients treated with the temsirolimus (mTOR inhibitor) and bevacizumab (VEGF inhibitor) combination were used in this study and in particular, treated versus non-treated melanoma tissues were analyzed. Wagenseller et al. found 12 miRNAs with tumor suppressor functions able to target 15 different oncogenes [98]. Among them, miR-125b, miR-7b, and miR-29c were identified as differentially expressed after the temsirolimus and bevacizumab combined treatment [164]. 

MiR-514a, a well-known key player in initiating melanocyte transformation and enhancing melanoma growth, has been reported to regulate the sensitivity of BRAF-targeted therapy by modulating the tumor suppressor NF1 gene [170]. In addition, it has been observed that miR-32 replacement therapy as a single agent is able to exhibit synergistic effects with vemurafenib [161]. 

These findings highlighted the important role of miRNAs in new therapeutic strategies that seek to overcome resistance. Different studies have demonstrated miRNAs’ impact in melanoma resistance to BRAFi, in ICIs resistance therapy, and in pairing therapies. It could provide even more of a basis for further studies against melanoma through miRNAs, which could represent attractive candidates for melanoma intervention [101,164,170].

### 2.2. MicroRNAs in Melanoma Immunotherapy 

The immune system plays a pivotal role in melanoma therapy, and specific immune therapies have been developed, such as anti-CTLA4 and anti-PD1-based immune therapies. Even so, most patients (50–60%) treated with these agents do not have a durable response [69,172]. 

Cancer immunotherapy inhibits the proliferation and invasion of cancer cells by inducing or enhancing anti-tumor immune responses in different ways, active or passive. It is the fourth efficacious and safe therapeutic option in addition to surgery, radiotherapy, and chemotherapy. 

Currently, many clinical trials using immunotherapy have made remarkable achievements. Phase II–IV clinical trials demonstrated that the use of immune checkpoint blockers (ICBs) for the treatment of MM patients significantly prolonged progression-free survival and overall survival in these patients [173,174]. Hence, different ICBs have been approved by the FDA for clinical therapeutic protocols in multiple tumors. Ipilimumab has been approved for melanoma while nivolumab and pembrolizumab have been approved for not only melanoma but also other malignant tumors. Ipilimumab, an anti-CTLA-4 mAb, shows a better effect on survival improvement in melanoma patients carrying the BRAF wild type compared to radiotherapy and chemotherapy. 

Despite a significantly prolonged progression-free survival and overall survival being observed, the majority of these patients developed resistance within one year [175]. 

MiRNAs have been demonstrated to regulate target genes involved in tumorigenesis and the development of melanoma, functioning similarly to oncogenes or anti-oncogenes, and to also play an important role immunotherapy [176] (Table 3). 

Notably, miR-30b/miR-30d involvement has been demonstrated in the melanoma metastatic process but not in the classic EMT invasive pathways. In particular, it has been observed that miR-30d-mediated GALNT7 (GalNAc transferase) inhibition stimulates the expression of the immune-suppressive IL-10 cytokine, which in turn triggers an immune-suppressive microenvironment [131]. MiR-210 is among the hypoxia-induced miRNAs in melanoma cells and it impairs the susceptibility to T-cell lysis by tumor cells [185]. Similarly, miR-34a/c has been demonstrated to modulate innate immune responses in melanoma cells by regulating ULBP2 expression, a stress-induced ligand of NKG2D [181]. Liu et al. showed that TGFβ induces miR-494 expression in myeloid-derived suppressor cells (MDSCs), promoting the accumulation and functions of the tumor suppressor [187]. 

Furthermore, it has been reported that miR-155 is involved in novel mechanisms adopted from melanoma cells to escape immune surveillance. MiR-155 modulated the IL-1β-induced downregulation of endogenous MITF-M expression in melanoma cells [183]. Moreover, forced miR-200a expression has been demonstrated to suppress CDK6 expression in metastatic melanoma cells, a factor frequently found to be dysregulated in cutaneous melanoma. Recently, CDK4/6 inhibitors have shown promising anti-tumor properties in several cancer types, including melanoma, and miR-200a downregulation seems to correlate with disease progression and a higher number of lymph node metastases [184]. 

### 2.3. MicroRNAs as Biomarkers

Currently, among the biomarkers able to offer the potential to predict the risk of progression to metastatic disease states in melanoma, LDH (lactate dehydrogenase) and S100B (S100 family of calcium-binding proteins) have been identified. LDH is the only accepted serum prognostic biomarker for routine clinical use in melanoma patients [188,189]. Unfortunately, limitations of the above-mentioned biomarkers due to the lack of sufficient sensitivity and specificity have been identified. In fact, LDH is not a melanoma-specific enzyme. It is also associated with many other benign and malignant diseases [190]. By contrast, S100B shows a strong association with melanoma prognosis. It is highly specific, and its increased levels are detected in patients with advanced melanoma [189,190,191,192].

Many different cell types, including tumor cells, produce exosomes. It has been demonstrated that miRNAs are localized in these vesicles, protecting themselves by RNase activity and securing their integrity [193]. Since then, circulating miRNAs have emerged due to their great potential and ability to discriminate among diverse types of cancers, and their chemical stability and resistance to RNase activity has been highlighted [194].

Leidinger et al. were one of the first research groups to perform high-throughput screening techniques for diagnostic circulating miRNA biomarkers’ identification [195]. Since then, circulating exosomal miRNAs have been extensively analyzed for their utility as biomarkers in different tumors and disorders, including MM and have become a putative clinical and prognostic biomarker [196,197].

Circulating miRNAs have also been found in biopsies, peripheral blood circulation, and other body fluids, and the profile of circulating miRNAs was shown to be able distinguish tumors from normal tissues [198,199]. Furthermore, miRNAs have an adequate half-life in clinical samples, if correctly handled and stored, and are rapidly detectable in plasma by accurate, specific, and noninvasive methods [200]. 

Stark et al. analyzed 17 miRNAs enriched in melanoma compared to 34 other solid cancer cell lines. The sera of melanoma patients (stage III and IV) were analyzed and a subset of seven miRNAs (MELmiR-7 panel) was derived. Within the MELmiR-7 panel, some miRNAs were identified as being able to discriminate different melanoma stages with a better diagnostic score than the currently applied serological tests based on LDH and S100B, and with high sensitivity and specificity [169]. 

Furthermore, Margue et al. also performed a whole miRNAs array of serum samples from melanoma patients compared to healthy individuals. They observed that miR-211 was very discriminative for stage IV tumors versus healthy controls, while miR-16 was rather downregulated in contrast to the upregulation reported by Stark [201]. Successive studies from Kanemaru et al. confirmed the discriminative power of miR-221 levels for MM patients and they also found a correlation between miR-221 levels and tumor thickness [202]. Finally, they observed that miR-221 levels were reduced after tumor surgical removal while they were increased in tumor recurrence cases. 

As highlighted by Jarry et al., limitations due to the use of different profiling platforms or variable techniques for serum and plasma preparation, RNA extraction, low concentration of secreted miRNAs, quality control, normalization, and statistical evaluation have been reported in the results of different studies on circulating miRNAs in oncology [203].

Mumford et al. also identified the potential of prognostic circulating miRNAs found to be differentially expressed in the circulation of melanoma patients compared to healthy controls, highlighting the technical variables that may lead to the lack of consistency between studies [204]. 

More than 500 miRNAs, identified in multiple studies, are present at higher levels in nevi or in melanomas. However, most of them have not been reproduced by using independent validation sets. Torres and colleagues refined this list down to six miRNAs potentially able to reproducibly distinguish nevi from melanoma across independent datasets and miRNA profiling platforms. Two miRNAs were highly expressed in melanomas (miR-31 5p, miR-21 5p) while four miRNAs (miR-211 5p, miR-125a 5p, miR-125b 5p, and miR-100 5p) showed decreased expression in melanoma. Among them, they confirmed the differential expression of miR-211 5p, miR-21 5p, and miR-125b 5p that has been previously linked to melanoma [205].

The importance of miRNA discovery as biomarkers in order to rapidly identify melanoma progression in patients would therefore be a significant clinical tool also because the limits of detection of diagnostic imaging (e.g., CT (Computed Tomography) scan, PET (Positron Emission Tomography) and MRI (Magnetic Resonance Imaging) that is not able to detect lesions that are <10 mm or low metabolic activity. Furthermore, no scans are sensitive enough to detect micro-metastases. The utility of miRNAs should be considered as a diagnostic and prognostic aid in the early detection of melanoma.

## 3. The Dual Role of miRNAs in Cancer

It has been widely demonstrated that miRNAs are able to modulate the expression of multiple targets, some of which play oncogenic or tumor-suppressive roles. Evidence suggests that some miRNAs can also have opposite effects in different tumoral contexts, as listed in Table 4.

Notably, the dual role of miRs and the melanoma system has been established, as summarized in Table 4. For instance, among the tumor suppressors, Skouti et al. showed miR-205-5p gradually decreased during melanomagenesis in mice and was able to reduce cell invasiveness and proliferation, and delay tumor initiation [216]. 

Several studies have focused on miR-205-5p and its dual role in cancer. It has been reported as oncomiR in lung and nasopharyngeal cancers by targeting PTEN [217,218,226]. Furthermore, a tumor suppressor role has also been described in prostate [219], breast [220], melanoma [227], glioblastoma [228], and colon cancers [229] by targeting c-MYC [230], PKCε [219], and VEGF-A [228]. Further, miR-9 has been found to be downregulated in metastatic melanomas compared to primary tumors. It has been shown to be able to downregulate SNAIL1 and consequently promote CDH1 expression, inhibiting melanoma cells’ ability to invade [72] while miR-9 has been described either as an oncomiR or tumor suppressor in a variety of other cancers [206]. 

MiR-21 negatively regulates MKK3 and acts as a tumor suppressor in melanoma by inhibiting cell growth and metastasis [207]. Instead, miR-21 inhibits tumor apoptosis and promotes proliferation and metastasis by downregulating p53 expression in uveal melanoma cell lines [208]. miR-125b represents another example of a miRNA able to act as either an oncomiR or a tumor suppressor, depending on the context. It acts as an oncomiR in the vast majority of hematologic malignancies but as a tumor suppressor in many solid tumors. This apparent paradox can be explained by considering the fact that a single miR-125b targets antiapoptotic factors (MCL1, BCL2L2, and BCL2), proapoptotic factors (TP53, BAK1, BMF, BBC3, and MAPK14), proproliferative factors (JUN, STAT3, E2F3, IL6R, and ERBB2/3), metastasis promoters (MMP13, LIN28B, and ARID3B), and metastasis inhibitors. 

MiR-125b has been found to be upregulated in some tumor types, e.g., colon cancer and hematopoietic tumors, where it displays an oncogenic potential, by inducing cell growth and proliferation and blocking apoptosis. In contrast, it acts in other tumor entities, e.g., melanoma, as a tumor suppressor by targeting c-Jun [96,213].

Indeed, miR-155 shows a dual role in various types of cancer cells, such as melanoma. Although miR-155 has been described as an oncogene in various type of cancers, Levati and colleagues demonstrated that miR-155 is able to inhibit the proliferation of melanoma cell lines by targeting the oncongene SKI [97]. Similarly, Li and colleagues and Qin and colleagues demonstrated that miR-155 exerts a tumor-suppressive effect in gastric cancer and ovarian cancer-initiating cells by targeting SMAD2 and CLDN1, respectively [215]. Another excellent example of the opposite roles is provided by miR-30d and miR-30b-5p, which are associated with progression from primary to metastatic melanoma [131].

MiR-30d acts as a tumor suppressor in prostate cancer cell proliferation and migration by targeting NT5E and is regulated by the Akt/FOXO pathway in renal cell carcinoma [211,212]. MiR-30b-5p acts as a tumor suppressor microRNA in esophageal squamous cell carcinoma [209]. MiR-30b suppresses tumor migration and invasion by targeting EIF5A2 expression in gastric cancer cells [210].

Furthermore, miR-146a has been shown to play a dual role in malignancy. MiR-146a has been identified as being able to promote the tumor growth of malignant melanoma and, at the same time, to impair tumor cell dissemination. High levels of miR-146a expression during melanoma progression triggers tumor growth through inhibition of lunatic fringe (LFNG) and NUMB and activation of the NOTCH/PTEN/AKT pathway. In contrast its downregulation in circulating tumor cells (CTCs) suppresses tumor dissemination through modulation of the expression of ITGAV and ROCK1 [98,99].

It has been shown that miR-211 exhibited a dual role in melanoma progression, promoting cell proliferation while inhibiting metastatic spread in a xenograft mice model [221].

High expression levels of miR-224-5p have been detected in a large variety of tumors, such as glioma, colorectal cancer, and renal carcinoma, and is downregulated in uveal melanoma. Notably, Li et al. showed that miR-224-5p is involved in the proliferation, invasion, and migration of uveal melanoma (UM) cells via regulation of the expression of PIK3R3 and AKT3 [110]. Results from Gan et al. highlighted the correlation of the downregulated expression of miR-224-5p with the clinical progression and prognosis of prostate cancer [222]. Knoll et al. showed that the miR-224/miR-452 cluster is significantly increased in advanced melanoma and that ectopic expression of miR-224/miR-452 induces EMT and cytoskeletal rearrangements, and enhances migration/invasion. Conversely, miR-224/miR-452 depletion in metastatic cells induces the reversal of EMT, inhibition of motility, loss of the invasive phenotype, and an absence of lung metastases in mice.

It has been shown that miR-224/miR-452 targets the metastasis suppressor TXNIP and induces feedback inhibition of E2F1. MiR-224/452-mediated downregulation of TXNIP is essential for E2F1-induced EMT and invasion [134]. Also, the tumor-suppressive role of miR-452 has been reported in gliomas, targeting stemness regulators, such as BMI-1 [230].

The Rang group’s results collectively indicated that miR-542-3p acts as a metastasis suppressor in melanoma [118] and as a tumor suppressor in ovarian cancer by directly targeting CDK14 ) and promoting the proliferation of osteosarcoma cells in vitro [231,232,233]. Furthermore, Haflidadóttir et al. reported miR148’s dual/opposite role in MITF regulation [84].

This set of observations highlights the polyvalence of miRNAs as an oncogenic or tumor suppressor, even within a single cancer type.

## 4. Conclusions

Until relatively recently, no viable treatment for metastatic melanoma patients had been detected. With the advent of target-based-therapy (BRAF and MEK inhibitors), immunotherapy (anti-PD1/PDL1 and anti-CTLA4 antibodies), and pairing therapies to avoid drug resistance, many improvements in progression-free survival have been achieved.

Scientific investigations have shown the involvement of miRNAs as a new key factor for melanoma metastasis treatment. Owing to their role in the regulation of gene expression and their stability (resistance to endogenous RNase activity) in body fluids, miRNAs have been extensively shown to be of particular interest for diagnosis, recurrence, identification, and treatment of cancer metastasis. Additionally, new techniques that are able to inhibit oncomiRs expression have been discovered and used as a new therapeutic option against many tumors: (1) Small molecule inhibitors (siRNA), (2) anti-miR oligonucleotides (AMOs), (3) miRNAs sponge, and (4) miRNA masking [234].

Although the limitations of miRNAs in target specificity have been shown, because they are able to regulate multiple ‘canonical’ instead of ‘non-canonical’ targets, there are currently clinical trials regarding the positive impact of miRNAs in different diseases. MiR-122/miravirsen (produced by Roche/Santaris) and miR-92/MRG 110 (produced by Regulus Therapeutics), designed to treat hepatitis C, are considered the flagship products of this class of future cancer drug development.

The increased target specificity and efficacy, and the minimization of side effects of miRNA drugs as intratumoral injections directly into the pathogenic site have been revealed for cancer-related pathologies [235,236]. In fact, Miragen has an active phase 1 study for miR-29 (MRG-201) to treat keloid and scar tissue formation as well as a phase 2 trial for miR-155 (Cobomarsen; MRG-106) for patients with a form of T-cell lymphoma.

Recently, the new miRNA drug candidate of Regulus (RGLS5579) targeting miR-10b has been developed for potential trials in glioblastoma multiforme patients, one of the most aggressive forms of brain cancer, with a median survival of approximately 14.6 months [237]. Therefore, the first recently completed phase 1 trial engaging a newer technology termed “targomiR” exhibited encouraging results in patients with recurrent malignant pleural mesothelioma or non-small cell lung cancer.

All of the miRNA-based drugs are currently in clinical trials and none have yet reached a pharmaceutical breakthrough. However, the acquisition of miRNA-based companies by major pharmaceuticals provides positive feedback regarding their potential.

In our opinion, despite the urgent need for the development of new and improved treatment strategies for melanoma patients, it will now be interesting to see how newer investigations of miRNAs will allow earlier detection of tumor recurrence and support the diagnosis and early detection of melanoma recurrence, as well as the prediction of patients’ outcomes/responses to therapies.

Hence, in this review, we highlighted the important role of miRNAs not only in the current treatments of melanoma metastasis but also their involvement in drug resistance to BRAF and MEK inhibitors, and their role as prognostic factors (biomarkers). This excitement for the positive role of miRNAs for melanoma metastasis treatment should encourage researchers, especially on the combinatorial approaches of miRNAs with the current pairing therapies, to increase the field of utilizing miRNAs as a therapeutic tool.

## Figures and Tables

**Figure 1 ijms-21-00878-f001:**
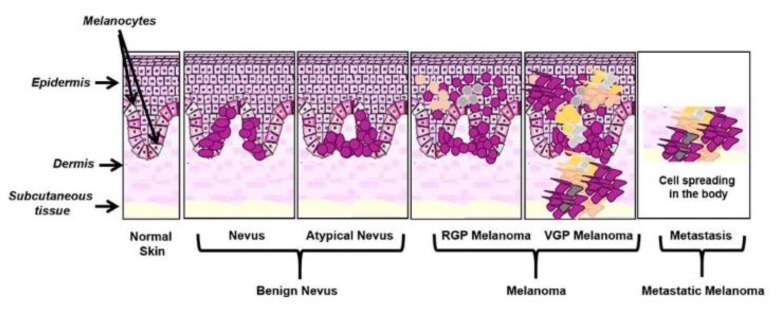
Schematic representation of melanoma onset and progression. Melanoma onset and progression described in the text was illustrated, underlining its clonal evolution, phenotype switching, and high heterogeneity [10].

**Figure 2 ijms-21-00878-f002:**
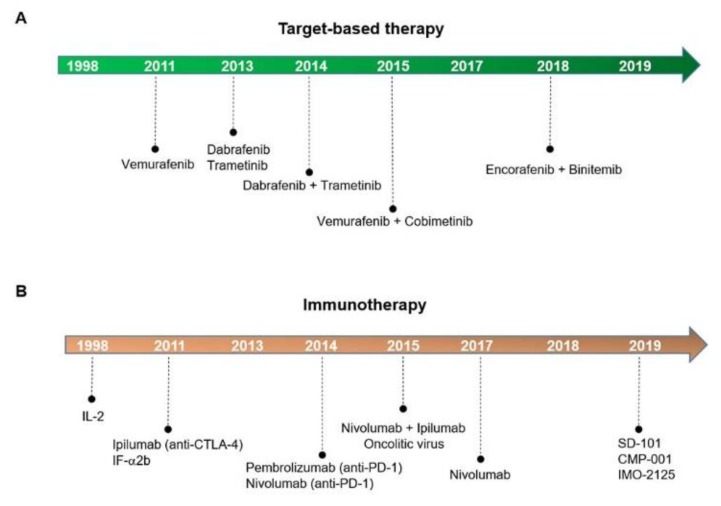
Melanoma treatment advances. Target-based therapy (**A**) and immunotherapy drugs (**B**) FDA-approved.

**Figure 3 ijms-21-00878-f003:**
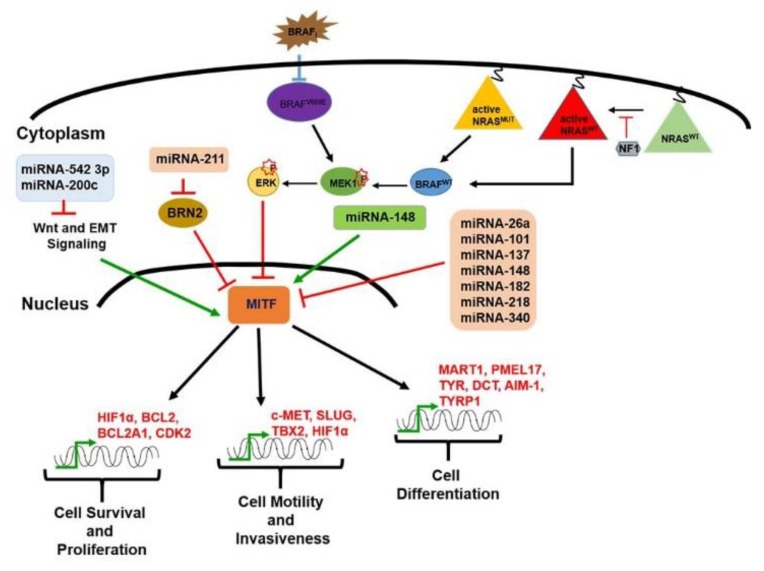
Schematic representation of several miRNAs able to regulate MITF, a master regulator of melanocyte development and of melanomagenesis.

**Table 1 ijms-21-00878-t001:** Most representative tumor suppressor miRNAs and OncomiR (orange) involved in melanoma metastasis.

miRNA	Function	Target	References
**miR-9**	Tumor suppressor	NF-κB1-SNAIL1	[74]
**miR-18b**	Tumor suppressor	MDM2	[75]
**miR-22**	Tumor suppressor	MMP14 and SNAIL	[76]
**miR-26a**	Tumor suppressor	MITF	[77]
**miR-34**	Tumor suppressor	c-Kit	[57]
**miR-30a 5p**	Tumor suppressor	SNAIL, Sox4	[78,79]
**miR-34b**	Tumor suppressor	MET	[80]
**miR-34c**	Tumor suppressor	MET	[80,81]
**miR-137**	Tumor suppressor	MITF; PIK3R3	[82,83]
**miR-148**	Tumor suppressor	MITF	[84]
**miR-145 5p**	Tumor suppressor	TLR4; Oct4, Sox2, c-Myc	[85]
**miR-138**	Tumor suppressor	HIF1α	[86,87]
**miR-150 5p**	Tumor suppressor	SIX-1	[88]
**miR-128**	Tumor suppressor	TERT	[89]
**miR-125a**	Tumor suppressor	Lin28B	[90]
**miR-193 b**	Tumor suppressor	CCND1	[91,92]
**miR-199 3p**	Tumor suppressor	MET	[93]
**miR-145 5p**	Tumor suppressor	TLR4	[94]
**miR-124**	Tumor suppressor	RLIP76	[95]
**miR-125b**	Tumor suppressor	C-jun	[96]
**miR-155**	Tumor suppressor	SKI	[97]
**miR-146a**	Tumor suppressor	ITGAV and ROCK1	[98,99]
**miR-194**	Tumor suppressor	GEF-H1/RhoA	[100]
**miR-199-3p**	Tumor suppressor	mTOR and c-Met	[101]
**miR- 200c**	Tumor suppressor	BMI-1	[102]
**miR- 205 5p**	Tumor suppressor	E2F1 and E2F5	[103]
**miR-211**	Tumor suppressor	AP1S2, SOX11, IGFBP5, SERINC3, RAP1A	[104]
**miR-203**	Tumor suppressor	BMI-1; SLUG	[105,106,107]
**miR-218**	Tumor suppressor	CIP2A, BMI-1, CREB1, MITF	[108,109]
**miR-224**	Tumor suppressor	PIK3R3/AKT3	[110]
**miR-365**	Tumor suppressor	NRP1	[111]
**miR-339 3p**	Tumor suppressor	MCL-1	[112]
**miR-338-3p**	Tumor suppressor	MACC1	[113]
**miR-340**	Tumor suppressor	MITF	[114]
**miR-339 3p**	Tumor suppressor	MCL1	[112]
**miR-429**	Tumor suppressor	AKT	[115]
**miR-579 3p**	Tumor suppressor	BRAF, MDM2	[116]
**miR-524 5p**	Tumor suppressor	BRAF, ERK2	[117]
**miR-542 3p**	Tumor suppressor	PIM1	[118]
**miR-605 5p**	Tumor suppressor	INPP4B	[119]
**miR-675**	Tumor suppressor	MTDH	[120]
**let7i**	Tumor suppressor	ITGB3	[121]
**let-7a**	Tumor suppressor	ITGB3	[122]
**let-7b**	Tumor suppressor	BSG; Cyclin D1/D3	[121,122]
**miR-10b**	OncomiR	ITCH	[123]
**miR-17**	OncomiR	ETV1	[124]
**miR-19**	OncomiR	PITX1	[125]
**miR-21**	OncomiR	TIMP3, PTEN, PDCD4, FBXO11; TP53	[126,127,128]
**miR-25**	OncomiR	DKK3; RBM47	[129,130]
**miR-30d**	OncomiR	GALNT7	[131]
**miR-30b**	OncomiR	GALNT7	[131]
**miR-125b**	OncomiR	NEDD9	[132]
**miR-146a**	OncomiR	NUMB	[99]
**miR-182**	OncomiR	MITF, FOXO3, MTSS1	[133]
**miR-214**	OncomiR	TFAP2C	[134]
**miR-224**	OncomiR	TXNIP	[135]
**miR-199a 5p**	OncomiR	ApoE; DNAJA4	[136]
**miR-199a 3p**	OncomiR	ApoE; DNAJA4	[136]
**miR-221**	OncomiR	c-KIT, P27KIP1	[137,138,139]
**miR-222**	OncomiR	c-KIT, P27KIP1	[137,138,139]
**miR-340**	OncomiR	MITF	[114]
**miR-373**	OncomiR	SIK1	[140]
**miR-452**	OncomiR	TXNIP	[135]
**miR-519d**	OncomiR	EphA4	[141]
**miR-532 5p**	OncomiR	RUNX3	[142]
**miR-638**	OncomiR	TP53, INP2	[143]
**miR-1908**	OncomiR	ApoE; DNAJA4	[136]

**Table 2 ijms-21-00878-t002:** The most representative miRNAs that are drug sensitive (green) and drug resistant (orange) to target-based therapy and chemotherapy in melanoma metastasis.

miRNA	Function	Target	References	Treatment
**miR-7**	Drug sensitive	EGFR/IGF-1R/CRAF	[159]	BRAFi
**miR-31**	Drug sensitive	SOX10	[160]	Chemotherapy
**miR-32**	Drug sensitive	MCL-1	[161]	BRAFi (vemurafenib)
**miR-126-3p**	Drug sensitive	ADAM9 and VEGF-A	[162]	BRAFi (dabrafenib)
**miR-199b 5p**	Drug sensitive	HIF-1a/VEGF	[163]	BRAFi
**miR-200c**	Drug sensitive	BM1	[157]	BRAFi
**miR-524 5p**	Drug sensitive	BRAF and ERK2	[117]	BRAFi
**miR-579 3p**	Drug sensitive	BRAF, MDMD2	[116]	BRAFi + MEKi
**miR-659 3p**	Drug sensitive	NFIX	[164]	chemotherapy (carboplatin/paclitaxel)
**miR-34a**	Drug resistant	CCL-2	[165]	BRAFi (vemurafenib)
**miR-30a 5p**	Drug resistant	IGF1R	[166]	Chemotherapy (Cisplatin)
**miR-100**	Drug resistant	CCL-2	[165]	BRAFi (vemurafenib)
**miR-125a**	Drug resistant	BAK1, MLK3	[167]	BRAFi
**miR-125b**	Drug resistant	CCL-2	[165]	BRAFi (vemurafenib)
**miR-204**	Drug resistant	NUAK1/ARK5, IGFBP5, TGF-bRII, Slug, and CHD5	[168,169]	BRAFi
**miR-211**	Drug resistant	NUAK1/ARK5, IGFBP5, TGF-bRII, Slug, and CHD5	[168,169]	BRAFi
**miR-514a**	Drug sensitive	NF1	[170]	BRAFi

**Table 3 ijms-21-00878-t003:** The most representative miRNAs involved in the regulation of melanoma immunotherapy.

miRNA	Target	References
**miR-17 5p**	ETV1	[177,178]
**miR-28**	TIM-3, B- and T-lymphocyte	[179,180]
**miR-30b/30d**	GalNac transferase	[132]
**miR-34a/c**	NKG2D, MICA/B; ULBP2	[181]
**miR-146a**	STAT1/IFN	[182]
**miR-155**	IL-1b, MITF-M	[183]
**miR-200a**	CDK6	[184]
**miR-210**	HIF-alpha	[185]
**miR-376**	MICB	[186]
**miR-433**	MICB	[186]
**miR-494**	PTEN	[187]

**Table 4 ijms-21-00878-t004:** The most representative miRNAs with an opposite role in melanoma and other tumors.

miRNA	References
miR-9	[75,206]
miR-21	[207,208]
MiR-30b	[209,210,211]
MiR-30d	[209,210,211,212]
miR-125b	[96,213]
miR-155	[97,214,215]
miR-146a	[98,99]
miR-205-5p	[216,217,218,219,220]
miR-211	[221]
miR-224 5p	[110,135,222]
miR-452	[108,135,223]
miR-542-3p	[118,224,225]

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
