# Peer review of "The miRNAs Role in Melanoma and in Its Resistance to Therapy"

_ijms, 2020, doi:10.3390/ijms21030878_

Round 1
Reviewer 1 Report
The review:” The miRNAs role in melanoma and in its resistance to therapy” is very extensive review.” Authors did a good job implementing available information about the miRNAs role in melanoma development.
The review:” The miRNAs role in melanoma and in its resistance to therapy” is very extensive review.” Authors did a good job implementing available information about the miRNAs role in melanoma development. Melanoma is a deadly type of skin cancer and the proper therapy is not yet well defined. There are many different approaches to melanoma treatment, but none of them had high success rate in treatment of high-grade melanoma. Different drug approach to melanoma treatment has been employed, but high resistance rate to this therapy makes is more complex. In recent years melanoma-associated, miRNAs emerged as possible therapy, which authors have described in this review. Authors nicely described current melanoma treatment modalities in details and pros and cons of each method.
Author Response
Thank you for your comments. We appreciate that our goals in writing this review were reached.
Reviewer 2 Report
this manuscript describes the role of miRNA in melanoma and its therapeutic potential.
The review is well organized and comprehensive
Minor comments:
p3, l 76: the melanocytes are resistance to UV-induced apoptosis and not its damage (they actually accumulate more mutations due to the exposure)
Circadian rhythm has been reported as a key risk factor in melanoma Generation. the readers will benefit if this topic will be integrated (as a risk factor and gene regulation of miRNA)
Author Response
Response to Reviewer 2.
Comments and Suggestions for Authors
this manuscript describes the role of miRNA in melanoma and its therapeutic potential.
The review is well organized and comprehensive
Minor comments:
p3, l 76: the melanocytes are resistance to UV-induced apoptosis and not its damage (they actually accumulate more mutations due to the exposure)
Thank you for your notice. We clarify this point by modifying the text in: The melanocytes are resistant to UV-induced apoptosis. They continue to grow and to accumulate genetic mutations leading to the formation and growth of a melanoma [22] [23].
Circadian rhythm has been reported as a key risk factor in melanoma Generation. the readers will benefit if this topic will be integrated (as a risk factor and gene regulation of miRNA).
Thank you for your suggestion. We integrated this point in Melanoma Risk Factors Section p.3, lane 80 adding a sentence: A role of circadian rhythm in melanoma development has been also reported. It has been demonstrated that melatonin, a major output product of circadian rhythm, play a protective role in melanoma, with the corresponding reference: Markova-Car EP1, Jurišić D, Ilić N, Kraljević Pavelić S. Running for time: circadian rhythms and melanoma. Tumour Biol. 2014; 35(9):8359-8368.
The reader could benefit from the cited review, more detailed and specific for this topic, which was not one of ours in our manuscript.